# Epigenetic Regulation in Uterine Fibroids—The Role of Ten-Eleven Translocation Enzymes and Their Potential Therapeutic Application

**DOI:** 10.3390/ijms23052720

**Published:** 2022-02-28

**Authors:** Marta Włodarczyk, Grażyna Nowicka, Michał Ciebiera, Mohamed Ali, Qiwei Yang, Ayman Al-Hendy

**Affiliations:** 1Department of Biochemistry and Pharmacogenomics, Faculty of Pharmacy, Medical University of Warsaw, Banacha 1B, 02-097 Warsaw, Poland; gnowicka@wum.edu.pl; 2Centre for Preclinical Research, Medical University of Warsaw, Banacha 1B, 02-097 Warsaw, Poland; 3The Center of Postgraduate Medical Education, Second Department of Obstetrics and Gynecology, 01-809 Warsaw, Poland; michal.ciebiera@gmail.com; 4Clinical Pharmacy Department, Faculty of Pharmacy, Ain Shams University, Cairo 11566, Egypt; mohamed.aboouf@pharma.asu.edu.eg; 5Department of Obstetrics and Gynecology, University of Chicago, Chicago, IL 60637, USA; yangq@bsd.uchicago.edu (Q.Y.); aalhendy@bsd.uchicago.edu (A.A.-H.)

**Keywords:** uterine fibroids, 5-hydroxymethylocytosine, TET enzymes, DNA demethylation, DNA methylation, epigenetic regulation, non-hormonal therapy

## Abstract

Uterine fibroids (UFs) are monoclonal, benign tumors that contain abnormal smooth muscle cells and the accumulation of extracellular matrix (ECM). Although benign, UFs are a major source of gynecologic and reproductive dysfunction, ranging from menorrhagia and pelvic pain to infertility, recurrent miscarriage, and preterm labor. Many risk factors are involved in the pathogenesis of UFs via genetic and epigenetic mechanisms. The latter involving DNA methylation and demethylation reactions provide specific DNA methylation patterns that regulate gene expression. Active DNA demethylation reactions mediated by ten-eleven translocation proteins (TETs) and elevated levels of 5-hydroxymethylcytosine have been suggested to be involved in UF formation. This review paper summarizes the main findings regarding the function of TET enzymes and their activity dysregulation that may trigger the development of UFs. Understanding the role that epigenetics plays in the pathogenesis of UFs may possibly lead to a new type of pharmacological fertility-sparing treatment method.

## 1. Introduction

Uterine fibroids (UFs) are monoclonal tumors originating from a single, undifferentiated mesenchymal cell, and consist of muscle and connective tissue [1]. UFs are characterized by increased deposition of a disorganized extracellular matrix (ECM). They contain a large number of proteins such as collagen, proteoglycan, fibronectin and are often separated from the myometrium by a pseudocapsule, a structure that surrounds UFs [2].

UFs are the most common tumors of the female reproductive organ with the incidence in women of reproductive age ranging from 20–40% to 70% in selected populations. They may be asymptomatic or cause clinical symptoms such as pelvic pain, massive menstrual bleeding, infertility or adverse perinatal outcomes [3]. Despite the high prevalence and social and economic costs of UFs, there are only a few treatment options for both UFs and their associated symptoms [4]. The size and location of the lesions are factors that influence the development of symptoms and necessitate treatment and its selection. Important determinants influencing the choice of a suitable therapeutic method include the patient’s age and reproductive plans. The dominant method of treatment of UFs is hysterectomy, however this prevents further reproduction of the woman. Currently available non-invasive treatments include high-intensity focused ultrasound (HIFU) or UF embolization (UFE). Among pharmaceuticals, gonadotropin-releasing hormone analogs (GnRH) lead to the desensitization of receptors and cause menopause-like effects, which may reduce the size and the severity of clinical symptoms [5]. Furthermore, progesterone receptor modulators alleviate UF symptoms, but currently, their use is limited due to reported cases of drug-induced liver failure [6].

The factors involved in the pathogenesis of UFs include genetic predisposition, environmental factors e.g., polychlorinated biphenyls, organochlorines, phthalates, and pesticides), steroid hormones, growth factors, mechanical forces, hypoxia, and oxidative stress which may induce fibrotic processes (Figure 1) [7]. Excess ECM with high amounts of glycosaminoglycans, fibronectins, and disordered, interstitial collagens, result in stiffness of UF tissue and generation of mechanical stress, which dramatically impacts the biochemical and biological response of the cells. It is generally accepted that UFs are hormone-dependent tumors, and progesterone and estrogens play a key role in their formation [8,9]. In addition to the role of hormonal factors, growth factors such as tumor growth factor beta (TGF-β), vascular endothelial growth factor (VEGF), and fibroblast growth factor (FGF) are related to the processes of fibrosis. Furthermore, a number of studies indicate a significant contribution from chromosomal damage. Whole-genome sequencing has revealed that genetic aberrations in *MED12* (mediator complex subunit 12), *FH* (fumarate hydratase), *HMGA2* (high mobility group AT-hook 2) are related to UF development [10]. Multiple genetic alterations are associated with the recognized subtypes of UFs that may require a different therapeutic approach.

The adaptation of the human genome to changing environmental conditions is possible thanks to the dynamic regulation of the expression of individual genes as a result of epigenetic mechanisms such as DNA methylation, post-translational modification of histone proteins, or non-coding RNA. The specific DNA methylation pattern depends on the stage of cell development, type, and function of the cell, and is the result of the balance between methylation and demethylation reactions [11]. Maintenance of the DNA methylation level is critical to genome stability and changes in the DNA methylation profile of tumor cells compared to normal cells have been widely discussed. Specific enzymes named DNA methyltransferases (DNMTs) transfer methyl groups from S-adenosylmethionine to the cytosine in the DNA chain. The DNA methyltransferase family includes two enzymes essential for de novo methylation DNMT3a and DNMT3b, and DNMT1 that recognizes and methylates the nonmethylated daughter strand during DNA replication. The short stretches of palindromic DNA with the sequence “CpG” (CpG islands) are fragments of the genome where cytosine methylation occurs most often. The presence of a methylated cytosine can repress transcription by inhibiting the binding of transcription factors or may promote the binding of other transcriptional repressors, including histone-modifying proteins, such as histone deacetylases (HDACs). DNA methylation is the basic mechanism for silencing multiple repeats fragments in genes and for inactivating one of the X chromosomes, a process that occurs in female cells during development [12].

Maintaining a specific DNA methylation pattern depends not only on the effectiveness of the methylation reaction, but is also the result of the balance between passive and active DNA demethylation [13]. Passive demethylation occurs during replication when methyltransferases do not methylate the newly synthesized DNA resulting in a loss of DNA methylation sites. Demethylation can also appear as a result of enzymatic mechanisms that rely on an excision-repair pathway to replace methyl cytosine with cytosine. These mechanisms can promote global or gene-specific changes in methylation. Active demethylation produced by specific enzymes is independent of DNA replication. The key role is played by TETs (ten-eleven translocation proteins), that convert 5-methylcytosine (5-mC) to 5-hydroxymethylcytosine (5-hmC). TET proteins are also involved in regulating other epigenetic alterations, i.e., modification of histones by interaction with specific proteins responsible for this process and their recruitment to chromatin. In humans, TETs have been recognized as a commonly present genetic feature in cancers. In several types of myeloid tumors, including myelodysplastic syndromes, myeloproliferative tumors, chronic myelomonocytic leukemia, and acute myeloid leukemia, TET genes were deleted or mutated [14,15]. Although demethylation is an essential process in the development of the organism, the mechanism of DNA demethylation is less understood than that of DNA methylation. It has been reported that demethylation modulates transcriptional responses to hormones and is associated with cancer, cardiovascular, and other pathologies [16,17]. Aberrant epigenetic modifications and abnormalities in the binding of transcription factors to chromatin may be implicated in transcriptional dysregulation and mutations [18]. Epigenetic processes are heritable and reversible, and therefore may be regulated by several exogenous and endogenous factors such as drugs, chronic stress, hormones, nutritional status, and exposure to environmental toxins [19].

TET-mediated epigenetic imbalance may lead to the discovery of new therapeutic targets in UFs. This paper presents an overview of the role of TET enzymes in the epigenetic context of UF development and their therapeutic potential.

## 2. Aberrant DNA Methylation in Uterine Fibroids

The involvement of abnormal DNA methylation in the pathogenesis of benign tumors such as UFs is widely accepted. Global genome hypomethylation may lead to the activation of proto-oncogenes [20]. Hypermethylation of the CpG island in the promoter regions of suppressor genes and genes involved in DNA repair has been associated with the silencing of their expression, causing a decrease in genome stability and inducing the growth of tumor cells [21].

It has been suggested that both methylation and demethylation play important roles in the pathogenesis of benign tumors such as UFs. Aberrant methylation of different gene regions, in which products are involved in various metabolic pathways, has been recognized [22,23,24]. Li et al., reported imbalanced methylation status in uterine tissue, which could be involved in UF development. In this study 10 methylated and 19 demethylated specific locations in UF tissue in comparison to myometrium were recognized [24]. Maekawa et al., demonstrated around 120 genes differently methylated in UFs compared to normal myometrium [25]. One of the proposed causes of the changes in methylation patterns was the aberrant activity of DNA methylation enzymes. Abnormal expression of *DNMT* genes was found in UFs [26]. mRNA expression of *DNMT3a* and *DNMT3b* was lower in UFs compared to the myometrium, whereas the level of *DNMT1* was equal or increased. In another study, the higher gene expression of *DNMT1* and *DNMT3a* proteins was seen in UFs compared to myometrium, and no significant difference in *DNMT3b* mRNA expression between UFs and myometrium were noted [24]. An increased *DNMT1* expression may reflect an elevated proliferative activity of UF cells as DNMT1 is responsible for the maintenance of DNA methylation patterns [27]. Inconsistent data from studies conducted so far may result from racial differences between participants, as well as methodological differences in assessing the level of gene methylation.

The balance between progesterone and estrogen action is essential for proper functioning of the uterus. Expression of human estrogen receptor 1 gene (*ESR1*), which codes for estrogen receptor-α (ER-α), is tissue specific. An aberrant DNA methylation of the *ESR1* gene in UFs was revealed [23]. DNA methylation status of the *ESR1* promoter region varied among individuals, reflecting personal physiological changes in uterine smooth muscle cells. Additionally, factors such as aging, chronic inflammation, or infection can influence the patterns of DNA methylation in the myometrium [28]. Progesterone receptor (PR) participates in the regulation of fertility, the development of mammary glands, and promotes the proliferation of the uterine epithelium. Progesterone activates smooth muscle cell division in the uterus, especially in the second phase of the menstrual cycle. Nowadays, progesterone is believed to be the main factor initiating uterine smooth muscle cell differentiation and abnormal growth [29]. Progesterone stimulates the development of UFs via PR and the PI3K/AKT signaling cascade [30,31,32]. However, the expression of PR by uterine cells is induced by estrogens via the estrogen-α receptor and consequently, the response to progesterone is dependent on the presence of an estrogen drive [33]. Conversely, *ESR1* expression in uterine cells is inhibited by progesterone through PR [34]. Additionally, there are several tissue-specific methylation sites around the PR locus [35].

Hormonal participation in the alteration of the methylation profile was widely studied in endometriosis, a disorder of the female reproductive tract which, similar to UFs, is also related to an estrogen and progesterone imbalance [36,37,38]. Promoter hypermethylation of genes associated with estrogen metabolism resulted in its downregulation in endometriosis [39,40,41]. Whereas, upregulation of genes associated with estrogen biosynthesis was related to the promoter’s hypomethylation [42,43,44,45]. In the endometrium of submucosal UFs, and intramural UFs, there were hypermethylated sites within the *HOXA10* gene compared to controls, whereas in women with endometriosis *HOXA10* gene was hypomethylated [46]. HOXA10 is a member of the abdominal B-related subclass of homeobox genes responsible for uterine homeostasis during development. *HOXA10* expression is regulated by estrogen and progesterone in the human endometrium and is involved in implantation. Therefore, alterations in gene methylation patterns regulated by hormonal changes may influence UFs fertility related problems [47].

Dysregulated DNA methylation mechanisms in UF stem cells compared to normal myometrium were also reported [48,49]. In UF stem cells, genes crucial for their differentiation were hypermethylated [50]. Furthermore, a recently published paper confirmed that the differentiation process of UF stem cells is linked to both progesterone signaling and DNA methylation [35]. In UF stem cells, endometrial cancer and endometriosis PR expression was inhibited due to the hypermethylation of the PR gene [40,51]. Progesterone receptor gene hypermethylation results in decreased expression of genes critical for progesterone-induced UF stem cell differentiation. PR knockdown was associated with increased global DNA methylation by regulating TET enzymes [35].

Moreover, in UFs, hypermethylation of tumor suppressor genes, including Kruppel-like factor 11 (*KLF11*), deleted in lung and esophageal cancer 1 (*DLEC1*), keratin 19 (*KRT19*), and death-associated protein kinase 1 (*DAPK1*) was recognized [52]. Additionally, *SATB2* (Special AT-rich sequence-binding protein 2) and *NRG1* (Neuregulin 1) were confirmed to be hypermethylated in UFs [23]. SATB2 is involved in regulation of chromatin structural organization, and as a transcription factor affects expression of a broad variety of genes. *SATB2* expression is associated with various types of cancers such as colorectal, head and neck, bone, breast, and pancreatic. SATB2 overexpression in UF cells was related to changes in cell morphology, cell adhesion, and aggregation [23,53]. Neuregulin-1 (NRG1), a member of the epidermal growth factor family, stimulates the proliferation, differentiation and survival of several cell types. NRG1 is also suggested to be involved in the development of various cancers [54]. SATB2 and NRG1 are engaged in the activation of WNT/β-catenin, transforming growth factor-beta (TGF-β), vascular endothelial growth factor (VEGF), platelet-derived growth factor (PDGF), epidermal growth factor (EGF), insulin-like growth factor-I (IGF-1) signaling pathways which are related to the pathogenesis of UFs [23]. Gloudemans et al., observed that in malignant smooth muscle tumors (leiomyosarcomas), the overall methylation of the *IGF-2* gene was low or absent and related to increased gene expression, while in normal smooth muscle tissues (myometrium) and benign smooth muscle tumors (leiomyomas) *IGF-2* was hypermethylated [55].

E-cadherin is a cell–cell adhesion molecule that regulate epithelial differentiation and proliferation. E-cadherin plays a role as tumor suppressor protein, and the loss of its expression in association with the epithelial mesenchymal transition occurs frequently during tumor metastasis. The methylation status of the E-cadherin gene (*CDH1*) promoter region in patients with UFs was related to E-cadherin expression. These results indicate a possible impact of epigenetic aberration on E-cadherin protein expression in endometrial tissue [56].

The main genetic alterations occurring in UFs are related to *MED12*, *HMGA1*, or *HMGA2* genes [57,58,59]. HMGA belongs to the non-histone family of chromatin-binding proteins that modify the conformation of DNA and thus the availability of DNA-binding proteins and influences the transcription of a variety of genes involved in cell growth, proliferation, and cell death [60]. Among the examined UFs, approximately 30–40% showed karyotypic abnormalities, with the most common being translocations in the 12q15 and 6q21 chromosome regions, leading to overexpression of *HMGA2* and *HMGA1*, respectively [61]. In healthy myometrium, the *HMGA2* gene is not highly expressed, but its overexpression is commonly seen in UFs [62]. Studies on *HMGA2* gene expression in UFs demonstrated its significant relation to hypomethylation in the *HMGA2* gene [63]. The hypomethylation of the *HMGA2* gene, along with the participation of the Let-7, family of microRNAs, are suggested mechanisms involved in the upregulation of *HMGA2* in myomas [64].

Receptor activator of nuclear factor κB ligand (RANKL), a progesterone/PR target gene, plays an important function in tumorigenesis. This pathway is believed to play a key role in the pathogenesis of UFs as blocking the RANKL/RANK pathway inhibits steroid hormone-mediated UF growth as demonstrated in a mouse xenograft model [65]. Methylation in the *MED12* gene was associated with altered expression of the *RANKL* gene via progesterone and progesterone receptors and resulted in the enhanced proliferation of stem cells and the development of a fibrous tumor [66].

## 3. TET Proteins-Characteristics and Function

In humans three isoforms of TET protein similar in structure and function are encoded by three different genes: *TET1*, *TET2* and *TET3*. The TET1 gene is located on human chromosome 10q21.3, TET2 on chromosome 4q24, and TET3 on chromosome 2p13.1 The TET family proteins belong to the dioxygenases and catalyze consecutive oxidation reactions to convert 5-methylcytosine into 5-hydroxymethylcytosine, 5-formylcytosine and 5-carboxylcytosine [67,68]. All TET proteins have identical catalytic activity to oxidize the 5-mC methyl group (Figure 2).

TET family proteins contain conserved catalytic domain located in the C-terminal region (Cys-rich domain), and the DNA recognition fragment in the N-terminal region- named the double-stranded β-helix (DSBH) domain [69,70]. The Cys-rich domain is comprised of two subdomains and modulates the chromatin targeting of TET proteins. The DSBH domain contains a His-X-Asp/Glu signature motif, a C-terminal conserved His residue and Arg residue that are involved in binding Fe^2+^ or 2-oxoglutarate respectively. [68]. At the N-terminus of the TET1 and TET3 proteins is located the CXXC domain with a zinc finger structure [70]. This fragment is responsible for binding both unmodified and methylated or hydroxymethylated cytosines. It has been suggested that the core catalytic domain preferentially binds cytosines in a CpG region but does not interact with surrounding DNA bases and shows specificity for flanking DNA sequences. In the structure of the members of the TET family there is also a spacer region that connects two parts of the disjointed enzymatic domain of DSBH. The C-terminal catalytic domain of TET2 comprises a Cys-rich and a double-stranded b helix (DSBH, also known as jelly-roll motif) domain. TET2 CXXC exists as a separate gene, also called IDAX or CXXC4, which encodes an inhibitor of Wnt signaling. IDAX is now believed to play a role in regulating TET2 activity by facilitating its recruitment to unmethylated CpG, although on the other hand it also lowers TET2 protein levels as a result of its caspase-mediated degradation activity [71]. TET enzymes are involved in the passive DNA demethylation that occurs during replication, when a lack of methylation appears in the newly synthesized DNA strand, which may be due to the presence of 5-hmC in the parental strand. As a result of the lower affinity of DMNT1 to 5-hmC than to 5-mC, the activity of DNMT1 is inhibited and unmodified cytosines appear in the daughter strand [72]. Therefore, a high concentration of 5-hmC in the DNA strand and the absence of functional de novo methylation activity (DNMT3a, DNMT3b) favors passive demethylation.

TET enzymes together with AID/APOBEC proteins are also involved in active DNA demethylation. The AID/APOBEC complex participates in catalyzing deamination of 5-mC and 5-hmC for thymine and 5-hydroxymethyluracil (5-hmU) [73]. High expression of the TET1 protein enhances demethylation by increasing the conversion of 5-mC to 5-hmC. The process of active demethylation of DNA depends on the availability of substrates and cofactors as well as post-transcriptional and post-translational regulation of TET and TDG (thymine DNA glycosylase). TDG participates in both DNA methylation and demethylation; therefore, *TDG* polymorphisms may be associated with genomic instability [74]. In *TDG* null embryonic stem cells five to ten fold increases in 5-fC and 5-caC levels were observed and in mice, embryonic deletion of *TDG* is lethal [75,76,77,78]. TDG and SLUG1 (single-stranded selective monofunctional uracil DNA glycosylase) remove thymine and 5-hmU from DNA and replace them with cytosine in a process called DNA base excision repair (BER) [79].

The TET family also interacts with many other proteins involved in the DNA repair process by base excision, such as MBD4 (Methyl-CpG Binding Domain 4, DNA Glycosylase), NEIL1, NEIL2, NEIL3 (nei endonuclease VIII-like), poly-ADP-ribose polymerase (PARP1, poly ADP-ribose polymerase 1), enzymes which recognizes single-strand breaks and LIG3 (DNA ligase 3) and XRCC1 (X-ray repair cross–complementing protein 1) involved in ligation of DNA after cytosine insertion [80,81]. This indicates that the TET-dependent 5-mC to 5-hmC oxidation followed by excision of oxidized cytosine by the BER repair system requires the participation of a large, coordinated protein complex.

The 5-hmC content varies between tissues [82,83]. High levels of 5-hmC have been found in embryonic stem cells (ESCs) and in the central nervous system. Low levels of 5-hmC were observed in liver, kidneys, large intestine and rectum, and the lowest in lungs, heart, breast and placenta [84]. Reduced levels of 5-hmC have been noticed in malignant neoplasms of the lung, colon, brain, breast, liver, prostate, kidney, melanoma, compared to healthy tissues [85]. This loss of 5-hmC in DNA strands may lead to hyper-methylation of tumor suppressor genes (TSG) frequently observed in cancers. Decreased 5-hmC levels were associated with poor prognosis and survival in breast cancer, laryngeal squamous cell carcinoma, renal, esophageal, gastric and hepatocellular carcinoma [86]. In UFs, a type of benign tumor, an increased amount of 5-hmC has also been reported [48]. As the observed 5-mC levels are strongly dependent on the activity of TET enzymes, gene mutations and the location of the TET proteins may play an important role. It was reported that p.H1802Q, p.H1802R and p.R1817S changes in *TET2* were associated with reduction in genomic 5-hmC levels in patients with diverse myeloid malignancies and other cancers [87,88,89,90]. Mutations in *TET2* linked with a decrease in 5-hmC levels were found in hematological cancers [91]. However, in UFs mutations or polymorphisms in *TET* genes have not been described.

Abnormalities in epigenetic regulation of gene activity influencing cell proliferation and differentiation underlie the development of various cancers. TET-catalyzed DNA hydroxymethylation plays a role as an enhancer in regulating chromatin accessibility to facilitate the genomic recruitment of transcription factors [71]. Therefore, understanding the mechanism of action of TET proteins and their participation in the formation of the epigenetic pattern of DNA is particularly important in the context of the pathogenesis and treatment of cancer. Data describing the relationship between the expression of individual proteins from the TET family and 5-hmC levels suggest that members of the TET family are not equally involved in 5-mC hydroxylation in different types of cancer. Ciesielski et al., showed decreased expression of *TET1* and *TET2*, and a positive correlation between 5-hmC levels and *TET1* and *TET2* expression but not TET3 in endometrial cancer [92]. Du et al., found that global 5-hmC levels were positively correlated with *TET1* expression but not with *TET2* and *TET3* in gastric cancer [93]. In melanoma and esophageal squamous cell carcinoma, decreased expression of *TET2* was found to be related to decreased levels of 5-hmC. It was suggested that all TETs catalyze the hydroxylation of 5-mC to 5-hmC, while only TET2 and TET3 are responsible for the subsequent removal of 5-hmC in the cytosine demethylation cascade [94,95,96]. However, the exact roles of the individual members of the TET family in regulating 5-hmC levels awaits clarification.

## 4. Role of TET Enzymes in UF Development

Available data on TETs and 5-hmC activity in UFs are sparse. Although more than 10 years have passed since the discovery of the TET enzymes, there are only a few studies exploring the role of TET proteins and 5-hmC levels in the development of UFs [48]. Elevated expressions of TET1 and TET3 but not TET2, have been detected in UFs compared with matched myometrium [97]. Significantly higher 5-hmC levels in UF tissue compared to normal myometrial tissue were recognized and increases in 5-hmC levels were associated with up-regulation of *TET1* or *TET3* mRNA and protein expression in UF tissue (Table 1). *TET1* or *TET3* knockdown significantly reduced 5-hmC levels in UF cells and decreased cell proliferation of primary UF cells. This indicates that an epigenetic imbalance in the 5-hmC content of UF tissue, caused by upregulation of the TET1 and TET3 enzymes, may be a new therapeutic target in UFs [97].

An increase in *TET1*, *TET3*, and *H19* expression in UF tissue samples compared to myometrium has been reported [99]. Cao et al., recognized that H19 promotes TGF-β signaling by upregulating expression of TGFBR2 and TSP1 via TET3-mediated epigenetic mechanisms. Furthermore, *H19* or *TET3* knockdown resulted in decreased expression of genes encoding collagenases: *COL3A1*, *COL4A1*, and *COL5A2*, and TGF-β pathway genes: *TGFBR2* and *TSP1*, indicating that H19 acts through TET3 to promote TGF-β signaling and ECM production [98,100].

Additionally, the importance of demethylation processes and the involvement of TETs in endometrial changes during the menstrual cycle, and impaired expression of *TETs* in endometrial cancer were described [92,101,102]. The expression of *TET1* and *TET3* was higher in the mid-secretory phase than in the other phases of the endometrial cycle. In in-vitro experiments, treatment of endometrial epithelial cells with progesterone induced *TET1*, *TET2*, and *TET3* expression, and estradiol plus progesterone treatment increased the expression of TET3. Experiments on stromal cells confirmed estradiol-induced *TET1* expression [63,103]. A study by Ciesielski et al., showed a decreased expression of *TET1* and *TET2* in endometrial cancer [92]. A positive correlation between 5-hmC levels and expression of *TET1* and *TET2*, but not *TET3*, was identified. Mutations in *TET2* associated with a decrease in 5-hmC levels were found in hematological cancers [91]. However, in UFs mutations or polymorphisms in *TET* genes have not been described. Summarizing the available data, the exact role of each TET family protein in the regulation of 5-hmC is still unclear, and further research is needed in UFs.

### 4.1. Factors Involved in the Regulation of TET Activity

Demethylation changes are gene-specific, they occur dynamically under various conditions, and there is both a loss and an increase in the level of 5-mC [104]. It is believed that the 5-mC content in the genome depends primarily on the activity of TET, which may be influenced not only by mutations in the encoding genes, but also by cofactors such as iron ions and α-KG and co-substrates like ascorbate and molecular oxygen (Figure 3) as TET enzymes are α-KG, oxygen- and iron-dependent dioxygenases [68,69].

#### 4.1.1. Alpha-Ketoglutarate

Alpha-ketoglutarate (α-KG) is an endogenous intermediary metabolite formed during the tricarboxylic acid cycle (TCA) by NADP-dependent isocitrate dehydrogenase isoforms (IDH1/2). Mutations in *IDH* genes result in production of modified proteins with the ability to synthesize 2-hydroxyglutarate (2-HG) from α-KG [105]. Due to its structural similarity to α-KG, 2-HG acts as a competitive inhibitor of α-KG -dependent dioxygenases including TET enzymes. Mutations in genes encoding two other enzymes involved in the Krebs cycle—succinate dehydrogenases (SDH) and fumarate hydratase (FH)—lead to the accumulation of succinate and fumarate that act as competitive inhibitors of TET enzymes [106,107,108] and, as a result, silence DNA demethylation [13,109,110,111].

The pathway analysis demonstrated significant dysregulation of TCA cycle metabolism in UF samples [112]. Among the TCA cycle metabolites fumarate, malate, α–ketoglutarate, and succinate were significantly increased in UFs of the FH subtype. Interestingly, fumarate, plays a role as a positive regulator of genome stability and fumarate accumulation promotes angiogenesis and cell proliferation [113].

Elevated 2-HG levels inhibited TET activity and induced histone and DNA hypermethylation in different models [114,115]. Experimental and clinical data support the hypothesis that 2-HG induced DNA methylation sustains stemness of cancer cells favoring their progression toward malignancy [116,117]. Heterozygous somatic mutations affecting *IDH1*/*IDH2* genes and DNA methylation profiles have also been found in glioma and acute myeloid leukemia (AML) [118]. Hypermethylation of many genes in glioblastoma as a result of mutations in the *IDH1*/*IDH2* genes and the inhibition of TET enzymes by 2-HG has been demonstrated [119]. In the study of Navaro et al., a significant reduction in 5-hmC levels in UF cells treated with 2-HG compared with vehicle-treated cells was observed [48]. It proves that 2-HG competitively inhibits TET activity in UF cells, leading to a reduction in 5-hmC levels and a significant decrease in cell proliferation. Therefore levels of 5-hmC may represent a new marker for the detection of growing UFs and 2-HG is worth being studied as a therapeutic target of UFs.

In in vitro cell experiments a decrease in genomic 5-mC accumulation was observed after α-KG supplementation [120,121]. These results suggest that exogenous α-KG can restore the activity of the TET enzyme inhibited by succinate that has accumulated in the cells and decrease DNA methylation levels [121]. Accumulation of α-KG due to α-ketoglutarate dehydrogenase inactivation triggers TET1 and TET3 protein expression and their enzymatic activity in breast cancer, limiting cell migration and epithelial–mesenchymal transition [119]. Treatment with exogenous α-KG was shown to induce a dose–dependent HIF-1α destabilization in hypoxia due to proteasomal degradation [122,123]. Moreover, a recent study showed that exogenous supplementation of α-KG is able to prevent tumor growth and metastasis formation of triple negative breast cancer cells by switching the metabolism from glycolytic to oxidative [124]. In these models, increased levels of α-KG trigger succinate dehydrogenase and fumarate hydratase levels while switching off glycolytic enzymes, inducing a decrease in fumarate and succinate abundance leading to HIF-1α destabilization. Upon treatment with α-KG derivatives, HIF downstream targets were downregulated, preventing neo-angiogenesis, metabolic alterations and apoptosis in cancer cells [118,125,126]. α-KG was also found to have an anti-proliferative effect by preventing DNA synthesis and inducing a cell cycle arrest in G1 phase [127]. As a metabolic intermediate α-KG is supposed to present with a low toxicity in normal cells, therefore, it has been proposed to use the increase in intracellular α-KG levels as a target of a new anti-cancer strategy. 2-HG itself affects a broad array of α-KG-utilizing enzymes and hence is not a suitable drug candidate for a selective TET inhibitor [128].

#### 4.1.2. Iron

The iron excess in cells may contribute to the formation of free radicals, lipid peroxidation, DNA and protein damage, leading to carcinogenesis. Moreover, iron depletion has been shown to induce global changes in histone and DNA methylation in cancer cells without affecting the expression levels of histone and DNA methyltransferases or demethylases, suggesting an effect on their enzymatic activity [129]. Iron ion (II) is also an important modulator of TET activity. To our knowledge, there are no studies describing the effect of iron availability on TET activity in UF cells. Fe(II) deficiency was related to reduced amounts of 5-hmC in mouse embryos and treatment with a specific Fe(II) chelator (thiosemicarbazone -TSC24), caused the activity of TET enzymes to decrease [130]. Due to excessive menstrual bleeding and anemia associated with UFs, it can be hypothesized that iron deficiency, by inhibiting the activity of TET enzymes and DNA demethylation, promotes the development of UFs.

#### 4.1.3. Ascorbic Acid

Vitamin C (ascorbate) is an essential micronutrient with antioxidant activity. It has been suggested that vitamin C deficiency may accelerate oncogenesis [131]. Vitamin C is required to maintain proper activity of TET enzymes. TET proteins contain iron (II) binding sites in the carboxyl terminus, which constitute their dioxygenase catalytic domain. Ascorbate is needed to reduce Fe^3+^ to Fe^2+^ and to allow binding of iron (II) ions to the C-terminal domain of TET [132,133]. The ability of vitamin C to modulate the activity of histone and DNA demethylating enzymes indicates that vitamin C administration can be useful in the treatment of epigenetic dysregulation by targeting aberrant histone and DNA methylation patterns associated with cancer progression [134].

Vitamin C treatment has been shown to significantly increase 5-hmC, 5-fC and 5-caC production in ESC and improve TET-dependent reprogramming of mouse and human fibroblasts [135,136,137,138]. It was reported that vitamin C increases TET2 enzymatic activity and promotes 5-hmC formation and DNA demethylation in MDS and AML cells [139]. In murine models of leukemia, administration of vitamin C has been shown to restore TET2 function, resulting in an increase in 5-mC formation, global DNA hypomethylation, self-renewal block, and inhibition of disease progression in *TET2*-deficient mice. Vitamin C treatment was also tested in murine IDH1 mutant leukemic cells, showing TET2-dependent 5-hmC gain, 5-mC loss, and upregulation of gene expression that correlates with decreased self-renewal of leukemia stem cells and increased differentiation towards the mature marrow phenotype [140]. These results suggest that enhancing functional TET activity, even in the presence of inhibitory oncometabolites, could be sufficient to restore epigenetic cues of differentiation and remove aberrant DNA and histone hypermethylation profiles [111]. Vitamin C-induced genome DNA hypomethylation was also observed in human leukemia cell lines and was associated with increased TET2 activity [133,141]. Increased TET activity results in enhanced oxidation of 5-mC to 5fC and 5caC, which are recognized by the base excision repair (BER) mechanism, and facilitate active DNA demethylation [142,143]. In *TET1*/*2* silenced cells ascorbic acid did not affect 5-mC oxidation. It was found that the reduced concentration of ascorbic acid in cells with normal expression of the *TET* genes significantly decreased 5-hmC levels in the lungs, liver, and brain [135].

The administration of vitamin C was reported to reduce blood loss during abdominal myomectomy [144]. On the other hand, vitamin C deficiency was shown to activate TGF-β signaling and impair collagen synthesis in UFs [145,146]. Metabolomic analysis of UF tissues with the *MED12* mutation showed dysregulation of vitamin C metabolism, but ascorbate levels alone remained unchanged [112]. No significant association between vitamin C and UF development was reported [147]. Therefore, new human data are needed to support the hypothesis that the observed impact of ascorbic acid in experimental studies, on TET activity, can be used to develop a new strategy for UF treatment or prevention.

#### 4.1.4. Hypoxia

Chronic hypoxia arises as a result of tumor angiogenesis and the tissue’s low availability of oxygen. The conversion of 5-methylcytosine to 5-hydroxymethylcytosine by TET enzymes requires oxygen, indicating that tumor hypoxia can disturb the activity of TET proteins and enhance DNA methylation. The hypoxic state in cancer development is associated with angiogenesis, epithelial–mesenchymal transition (EMT), and metastasis [148]. As a stress factor, hypoxia enhanced epigenetic mechanisms such as hypomethylation of the *HMGA2* gene leads to overexpression of the *HMGA2* gene and induction of UF development [1]. However, conflicting data on changes in hypomethylation levels in neoplastic cell lines under hypoxia are available. Global 5-hmC levels were found to be decreased in cancer cell lines like neuroblastomas (N2A), and liver cancers (Hep3B) but they were increased in some N-type Neuroblastoma cell lines and breast cancer cell lines (4T1) [114,149]. The oxygen KM for the purified catalytic domains of TET1 and TET2 has been reported to be low indicating that the TETs can remain at least partially active under low oxygen conditions [150]. On the other hand, hypoxia can result in increased production of reactive oxygen species (ROS), which can affect ferrous ion availability for TET enzymatic activity [151]. In addition, under hypoxia, cellular concentrations of oncometabolites such as fumarate can affect TET activity [152].

#### 4.1.5. miRNA

MicroRNAs (miRNAs) are short (19–25 nt) non-coding RNAs that bind to complementary sequences within messenger RNA (mRNA) molecules. The miRNAs function is gene silencing via translational repression or target degradation. Additionally, a single mRNA can be modulated by multiple different miRNAs, resulting in a regulation of the complex gene network [153].

Several miRNAs such as let-7, 200a, 200c, 93, 21, 26a, 106b were identified to be involved in UF development by regulation of inflammation, cell proliferation, angiogenesis, apoptosis, and ECM synthesis [154,155,156,157]. Despite no clear evidence about a clear impact of miRNA on the activity of TET enzymes in UF development, the relationship between miR-129 and the target gene *TET1* was documented [158,159]. It was recognized that miR-129 can affect cell proliferation and apoptosis through regulating the expression of *TET1*, thus participating in the occurrence of UFs. Highly expressed miR-129 reduced TET1 protein and mRNA expression in UF cells [160]. MiR-129 expression was repressed by estrogen and progesterone, and its downregulation was beneficial to the development of UFs. These results suggest that further study of miR-129-TET1 and DNA demethylation in the apoptosis pathway will provide novel ideas for exploring the mechanism and treatment of UFs.

Among other microRNA molecules, miR-26a has been shown to bind to TET1, 2, 3 while miR-29b specifically targets TET1, resulting in a reduction in 5-hmC mediated demethylation during embryogenesis [161,162]. MiR-22, which targets all three TETs and reduces 5-hmC levels, especially in the loci of mir-200, promotes cancer progression and metastasis in breast cancer [163]. Over 30 miRNAs were reported to inhibit TET2, regulating hematopoiesis of *TET-2*-wild-type (without mutation) acute myeloid leukemia compared to *TET-2* mutant cells [164]. TET1 targeting miRNAs such as miR-520b, and miR-191 have been found to be elevated in intrahepatic cholangiocarcinoma, promoting proliferation, migration, and invasion [165,166]. In ovarian cancer cells, activity of DNMT1 was inhibited by miR-152 and miR-185 which are recognized as tumor suppressor miRNAs. In endometrial cancer miR-152 was identified as being silenced by DNA hypermethylation [167]. miR-29b promotes in vitro ESC differentiation through the TET1-mediated demethylation pathway. miR-29b reduces cellular 5hmC levels during ESC differentiation via suppression of TET1 gene expression, and regulates mesendoderm-specific differentiation of ESCs both in vitro and in vivo [168]. Further studies on the role of miRNA in activation of TET enzymes and UF development are needed.

## 5. Demethylating Agents and UF Development

Due to the reversible nature of epigenetic changes, various exogenous compounds are being investigated and proposed as new therapeutic agents in several cancers. Demethylating agents, like 5-azacytidine (5′-Aza), 5-aza-2′-deoxycytidine (decitabine, DAC) are nucleoside analogs of cytidine, which are incorporated into DNA during S-phase, and inhibit DNMT1 activity which leads to global hypomethylation [169]. Decitabine, a FDA-approved agent, is one of the most widely used DNA methyltransferase inhibitors, not only in hematological malignancies, but also in solid tumors. DAC treatment upregulates the expression of some tumor repressor genes that have been silenced because of promoter hypermethylation. However, DAC use is limited by its toxic effects [170,171,172].

Several studies also demonstrated an effect of demethylating agents on UFs. Treatment of a xenograft mouse model with both 5′-Aza, and an antiprogestational agent (mifepristone) resulted in an almost complete reduction in UF tumors [35]. Therefore, 5’-Aza, may activate PR signaling, and stimulate differentiation of UF stem cells to reduce tumor size in vivo. These findings reveal crosstalk between epigenetic and hormonal regulation during UF stem cell differentiation, and suggest a new treatment strategy. Moreover, in human UF primary cells, 5-aza, a DNMT inhibitor, reduced ECM formation and expression of c-MYC and MMP7, which are the final targets of the Wnt/β-catenin signaling pathway [173]. As a source of growth hormones and soluble profibrotic factors that promote tumor growth, the ECM is considered as a potential therapeutic target for UFs [174]. 5-aza-CdR treatment significantly decreased the expression of ECM and ECM-associated proteins such as fibronectin, collagen I, and PAI-1. These findings suggested an important role of 5-aza-CdR in the epigenetic regulation of key fibrotic proteins involved in UF formation. Shimeng Liu and coworkers, explored a way of reducing UF lesions through treatment by 5′-Aza as an epigenetic modulator [50]. Locus-specific genomic analyses revealed differential methylation of genes involved in cell differentiation/proliferation, such as at *ESR1*, associated with altered mRNA levels, indicating a clear impact of 5′-Aza treatment in inducing genome demethylation. [50,152]. A mouse model of xenografts of human UF explants under the renal capsule was treated with 5′-Aza and showed a massive regression in the tumor size (36% of the size observed in the controls) [29]. 5′-Aza treatment shifts the balance of genes towards cell differentiation by modifying and, as a result, mimicking the demethylation observed in this process. UF stem cells are associated with low TET1 transcripts levels and enhanced gene methylation. 5′-Aza treatment leading to gene demethylation resulted in slowing down the tumor growth by depleting stem cells. Treatment with 5’-Aza reduced the population of UF stem cells (LSC) by approximately 40%, while primary LM cells pretreated with 5′-Aza produced much smaller tumors [50]. Therefore, being an accepted epidrug, 5-Aza can be considered a part of UF therapies that slow tumor growth by depleting stem cells through specific demethylation reactions. Decitabine was found to increase 5-hmC at hemimethylated regions by TET enzymes in the mechanism of passive DNA demethylation. Combined treatment with vitamin C and decitabine decreased cell proliferation and increased cell apoptosis. This synergistic effect was reduced upon *TET2* knockdown in A2780 cells, demonstrating a key effect of TETs in treatment with demethylating agents [141,175]. Presented data indicate that DNMT inhibitors can be considered a new therapeutic option for UFs.

Chua at al., identified a promising cytosine-based compound, Bobcat339, that has inhibitory activity against TET1 and TET2, but does not inhibit the DNA methyltransferase, DNMT3a [176]. Singh et al., identified a specific TET inhibitor, which recognizes the catalytic core of TET enzymes and selectively interferes with their enzymatic activity [177]. Moreover, some fluorinated compounds (the 2′-(R)-fluorinated derivatives F-hmdC, F-fdC, and F-cadC) have been found to be substances that can be used to study active demethylation and are promising tools to investigate TET activators and inhibitors [178,179].

Unlike genetic changes which are permanent, epigenetic mechanisms are potentially reversible and thus are promising targets for anti-cancer therapy. According to the knowledge that about 70% of annotated gene promoters contain CpG-rich regions, they might be a target for epigenetic drugs. Future studies are needed to identify and characterize specific TET inhibitors or activators and their mechanisms of action. The administration of the compound directly to changed tissue and the surrounding area would avoid epigenetic changes in other tissues in the organism. One of the challenges is to target specific genes—the treatment causes hypermethylation at certain loci and hypomethylation at others. In addition, the biological significance and specific contributions of enzymatic and non-enzymatic activities of TET proteins remain largely unknown. Thus, more intensive research on compounds influencing TET enzyme activity is needed to find other epigenetic modulators with therapeutic effects on UFs that can reduce tumor formation.

## 6. Conclusions

Growing evidence highlights epigenetic mechanisms, especially interactions between TETs, active DNA demethylation, gene expression, and DNA damage response in UF development. DNA methylation and demethylation-dependent gene regulation may play a key role in the formation of UFs. However, it also implies that a network of various epigenetic factors is involved in the maintenance of DNA methylation in cells and tissues. Unfortunately, data on TET proteins expression and the levels of 5-hmC in UFs are limited. Therefore, deep mechanistic insights into the clinical relevance of various factors regulating DNA methylation–demethylation dynamics will lead to non-hormonal, fertility-friendly drugs for treating patients with this clinically significant disease.

## Figures and Tables

**Figure 1 ijms-23-02720-f001:**
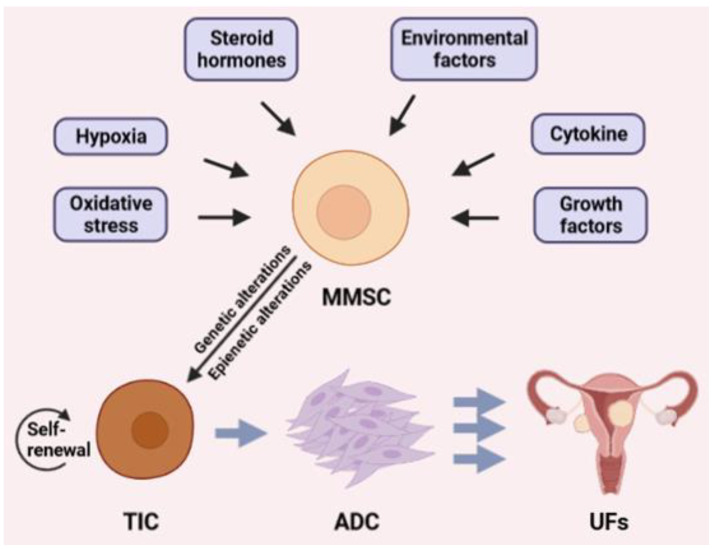
Several factors, including growth factors, oxidative stress, steroid hormones, hypoxia, environmental insults, and cytokines, may trigger the conversion of MMSC to TIC leading to the formation of UFs. MMSC: myometrial stem cell, TIC: tumor-imitating cell, ADC: abnormally differentiated cells.

**Figure 2 ijms-23-02720-f002:**
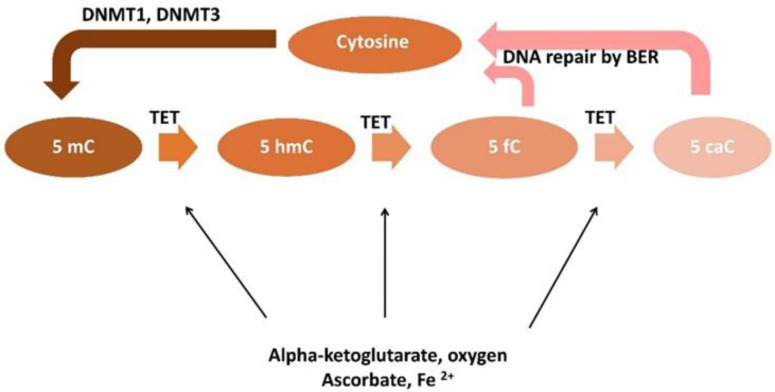
Cytosine transformations are catalyzed by the TET enzymes and their co-substrates (alpha-ketoglutarate, oxygen) and co-factors (ascorbate, Fe^2+^). TET: Ten-Eleven Translocation enzymes; DNMT1,3: DNA methyltransferases; BER: Base Excision Repair; 5-mC:5-methylcytosine, 5-hmC: 5-hydroxymethylcytosine; 5fC: 5-formylcytosine; 5caC: 5-carboxylcytosine.

**Figure 3 ijms-23-02720-f003:**
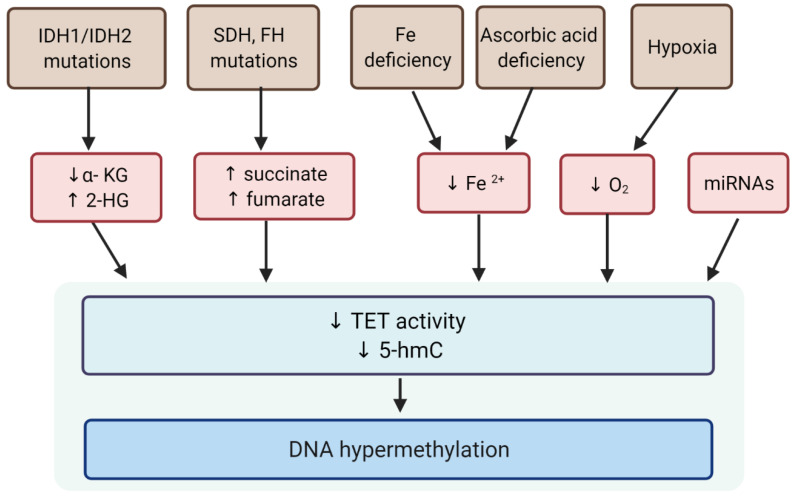
Factors involved in the regulation of TET activity. IDH1/IDH2: NADP-dependent isocitrate dehydrogenase; SDH: succinate dehydrogenases; FH: fumarate hydratase; α-KG: alpha-ketoglutarate; 2-HG: 2-hydroxyglutarate; TETs: Ten-eleven translocation enzymes; 5-hmC: 5-hydroxymethylcytosine.

**Table 1 ijms-23-02720-t001:** Overview of the TET-related studies in UFs.

Biological Samples	Results	References
UF tissues and matched myometrium Primary UF cell culture	*TET1* and *TET3* gene expression were higher in UF tissue and UF cells compared to normal myometriumNo difference in gene expression of *TET2*	[48]
Primary UF cell culture	Increased gene expression of *TET3* after estrogen and progesterone treatment	[98]
Primary UF cell culture ex vivo explant culture of UF tissue	Transcript levels of *TET1* and *TET3* were lower in UF stem cell-like cells and was related to global hypermethylation	[50]

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
