# Peer review of "Epigenetic Regulation in Uterine Fibroids—The Role of Ten-Eleven Translocation Enzymes and Their Potential Therapeutic Application"

_ijms, 2022, doi:10.3390/ijms23052720_

Round 1

Reviewer 1 Report

Through epigenetic mechanisms, TET enzymes play an important role in cell physiology and have implications in various pathologies. In this article, authors review TET chromatin biology and provide a succinct description of data that implicates TET enzymes in uterine fibroids. The authors provide a summary of their potential as a therapeutic target.

Overall, this is a well written article that provides a crisp description of TET function and a resource for navigating a complex topic. I have some suggestions that further focus specific topics as well as few minor edits.

Table 1 is not very informative. I would consider adding isoform information as well as TET domains and specific reported complexes with AID/APOBEC or other factors.

Italicize all gene names

Lines 70-71; statement unclear requires reference and explanation

Lines 80-81; awkward sentence

Line 107 should read “changes in the methylation”

Lines 114-115 states “It has been reported” yet there are no provided references of primary articles.

Lines 116-122 statements should more specifically be in the context of specific DNA de/methylation.

Line 127 gene should be “DNA’’

Line 142 DNMT already defined above

Line 153-154 ER gene should be specifically stated (ESR1)

Line 174 possessive for promoter should be re-written

Line 207 specify a smooth muscle malignancy

Lines 249-250 more context needed for overexpression in breast cancer study, similarly the next sentence states lack of TET. Please add clarify

BER not defined in figure 2 legend, nor is it appropriately referenced in the main text

Lines 283-286 statements missing period. Also, sentence can be modified/expanded to clarify TET DNA binding activity and specificity, including the following sentences about the CXXC domain, with primary references and a description of this data. It would also be appropriate to discuss the context of TET targeting to chromatin and whether this has been investigated in UF.

Line 289 remove domain hyphen; element not accurate – domain/motif?

Line 425 was this total amounts or at specific regions of the genome? A description of the data from these studies in this context is needed

Line 431 reference format

Lines 439-440 awkward sentence

Line 500 hypoxia section lacks reference to the above HIF1a findings with a-KG

Line 544 DNMT1 is already defined

Line 541 wild-type AML?

Line 548, TET1 repression? Transcriptional? The miRNA? Please clarify

Could use additional section in conclusions about challenges to TET therapies for UF

Through epigenetic mechanisms, TET enzymes play an important role in cell physiology and have implications in various pathologies. In this article, authors review TET chromatin biology and provide a succinct description of data that implicates TET enzymes in uterine fibroids. The authors provide a summary of their potential as a therapeutic target.

Overall, this is a well written article that provides a crisp description of TET function and a resource for navigating a complex topic. I have some suggestions that further focus specific topics as well as few minor edits.

Table 1 is not very informative. I would consider adding isoform information as well as TET domains and specific reported complexes with AID/APOBEC or other factors.

Italicize all gene names

Lines 70-71; statement unclear requires reference and explanation

Lines 80-81; awkward sentence

Line 107 should read “changes in the methylation”

Lines 114-115 states “It has been reported” yet there are no provided references of primary articles.

Lines 116-122 statements should more specifically be in the context of specific DNA de/methylation.

Line 127 gene should be “DNA’’

Line 142 DNMT already defined above

Line 153-154 ER gene should be specifically stated (ESR1)

Line 174 possessive for promoter should be re-written

Line 207 specify a smooth muscle malignancy

Lines 249-250 more context needed for overexpression in breast cancer study, similarly the next sentence states lack of TET. Please add clarify

BER not defined in figure 2 legend, nor is it appropriately referenced in the main text

Lines 283-286 statements missing period. Also, sentence can be modified/expanded to clarify TET DNA binding activity and specificity, including the following sentences about the CXXC domain, with primary references and a description of this data. It would also be appropriate to discuss the context of TET targeting to chromatin and whether this has been investigated in UF.

Line 289 remove domain hyphen; element not accurate – domain/motif?

Line 425 was this total amounts or at specific regions of the genome? A description of the data from these studies in this context is needed

Line 431 reference format

Lines 439-440 awkward sentence

Line 500 hypoxia section lacks reference to the above HIF1a findings with a-KG

Line 544 DNMT1 is already defined

Line 541 wild-type AML?

Line 548, TET1 repression? Transcriptional? The miRNA? Please clarify

Could use additional section in conclusions about challenges to TET therapies for UF

Author Response

  We  thank  the  reviewer  for  their  careful  reading  of  the  manuscript  and  their constructive remarks.  We  have  taken  the  comments  on  board  to  improve  and  clarify  the  manuscript.  Please  find below  a  detailed  point-by-point  response  to  all  comments  (reviewers’  comments  in  black,  our  replies  in blue).

Through epigenetic mechanisms, TET enzymes play an important role in cell physiology and have implications in various pathologies. In this article, authors review TET chromatin biology and provide a succinct description of data that implicates TET enzymes in uterine fibroids. The authors provide a summary of their potential as a therapeutic target.

Overall, this is a well written article that provides a crisp description of TET function and a resource for navigating a complex topic. I have some suggestions that further focus specific topics as well as few minor edits.

Table 1 is not very informative. I would consider adding isoform information as well as TET domains and specific reported complexes with AID/APOBEC or other factors.

Thank you for this comment. As the main axis of this article is the possible role of TET in the development of uterine sarcomas, we did not want to focus on the exact structure of these proteins. We delated Table 1 and added additional information about TET’s structure to the text.

Italicize all gene names

We corrected gene names.

 Lines 70-71; statement unclear requires reference and explanation

We decided to delete this sentence.

 Lines 80-81; awkward sentence

We decided to delete this sentence.

Line 107 should read “changes in the methylation”

We corrected the sentence.

Lines 114-115 states “It has been reported” yet there are no provided references of primary articles.

We added appropriate reference.

 Lines 116-122 statements should more specifically be in the context of specific DNA de/methylation.

We deleted the sentence.

Line 127 gene should be “DNA’’

We corrected the sentence.

Line 142 DNMT already defined above

We corrected the sentence.

Line 153-154 ER gene should be specifically stated (ESR1)

We corrected ER gene name to ESR1 where needed.

Line 174 possessive for promoter should be re-written

 We corrected the sentence.

Line 207 specify a smooth muscle malignancy

We clarify a smooth muscle malignancy.

 Lines 249-250 more context needed for overexpression in breast cancer study, similarly the next sentence states lack of TET. Please add clarify

As suggested by the second Reviewer, we have decided to delete the entire paragraph

BER not defined in figure 2 legend, nor is it appropriately referenced in the main text

We decided to delete Figure 2, as it didn’t contain any relevant information.

Lines 283-286 statements missing period. Also, sentence can be modified/expanded to clarify TET DNA binding activity and specificity, including the following sentences about the CXXC domain, with primary references and a description of this data. It would also be appropriate to discuss the context of TET targeting to chromatin and whether this has been investigated in UF.

We added additional information to the text.

Line 289 remove domain hyphen; element not accurate – domain/motif?

 We corrected the sentence.

Line 425 was this total amounts or at specific regions of the genome? A description of the data from these studies in this context is needed

We corrected the sentence.

Line 431 reference format

We corrected the reference.

Lines 439-440 awkward sentence

We corrected the sentence.

Line 500 hypoxia section lacks reference to the above HIF1a findings with a-KG

We added the reference.

 Line 544 DNMT1 is already defined

We corrected the sentence.

Line 541 wild-type AML?

We clarified the sentence.

Line 548, TET1 repression? Transcriptional? The miRNA? Please clarify

We clarified the sentence.

Could use additional section in conclusions about challenges to TET therapies for UF

We added a text fragment in the conclusion summarizing our opinion about the challenges to TET therapies for UF .

Reviewer 2 Report

WÅ‚odarczyk et al summarize the role of ten-eleven translocation (TET) proteins in the development of uterine leiomyomas. Fibroids have long been a challenge to clinicians and researchers, ands this article aims to join the discussion to place this problem in an epigenetic context. The submission has merit as a review article, but requires attention to several issues as discussed below.

Poor organization and lack of clarity reduce the quality of this work considerably. The confusion starts early, where the terms TET enzyme and TET protein are freely interchanged. Does the work claim that these are the same thing? What is a ‘non-hormonal, fertility friendly drug’ and how does an understanding of epigenetics lead to its use?

In general, abbreviations should not be in a manuscript title - not all readers will have familiarity with TET. While this abbreviation is defined halfway thru the Abstract, oddly the TET term drops out completely to receive zero attention in the Introduction. So what was the purpose of the introduction? Perhaps TET can be mentioned here as a common genetic feature in malignancy, a translocation involving chromosomes 10 and 11 .. creating a MLL-TET1 fusion protein (Lorsbach, 2003)?

Another example where coherent assembly is missing is the topic of DNA methylation, which is the main idea driving this project. The reader meets this subject on p.9, line 336 (just below Table 2) but then the exact same idea is picked-up again much later, as an entire section on p.13, line 552+. One wonders if six authors just independently gathered literature on their own, then put everything together later with no overall plan.

Minor points

See p.2, line 61: what ‘mechanical forces’ induce uterine fibroids?

See p.2, line 70-71: “…mutations are believed to be precursors to chromosomal rearrangements that alter genomic integrity …” Certainly a big insight in 1920, but not sure what this adds in the modern era.

See p.4, lines 154-158: “However, DNA methylation status…” This long passage does not appear to be a complete sentence.

See p.5, lines 237-241: It is unclear how endometrial function or menstrual cyclicity is relevant to myomas, except for cases where fibroids impinge into the uterine cavity. Indeed, if the process-overlap between UF and endometrium is so rare, then this part could be omitted entirely.

Figure 2 caption: Please define all abbreviations in the figure legend.

The authors may wish to consider developing a summary diagram (alongside Section 5.1, which is data-rich) to illustrate how proposed UF interventions work, either alone or in concert with other therapeutic modes.

Author Response

 We  thank  the  reviewer  for  their  careful  reading  of  the  manuscript  and  their constructive remarks.  We  have  taken  the  comments  on  board  to  improve  and  clarify  the  manuscript.  Please  find below  a  detailed  point-by-point  response  to  all  comments  (reviewers’  comments  in  black,  our  replies  in red).

WÅ‚odarczyk et al summarize the role of ten-eleven translocation (TET) proteins in the development of uterine leiomyomas. Fibroids have long been a challenge to clinicians and researchers, ands this article aims to join the discussion to place this problem in an epigenetic context. The submission has merit as a review article, but requires attention to several issues as discussed below.

Poor organization and lack of clarity reduce the quality of this work considerably. The confusion starts early, where the terms TET enzyme and TET protein are freely interchanged. Does the work claim that these are the same thing? What is a ‘non-hormonal, fertility friendly drug’ and how does an understanding of epigenetics lead to its use?

As suggested, we added the full name of the TET enzyme in the title of the manuscript. We also combined the Introduction with the Epigenetic Context section to provide an introduction to the meaning of DNA methylation and TET enzymes.

According to the knowing statement that not all proteins are enzymes, but most enzymes are proteins and that TET proteins have enzymatic activity and belong to the family of ten-eleven translocation (TET) methylcytosine dioxygenases we used terms TET protein and TET enzymes interchangeable.

Thank you for this important remark about “fertility friendly drug”. Drugs targeting demethylation or DNA methylation enzymes could have an impact on gene expression without involving biochemical pathways and known group of side effects in contrast to available hormonal therapy (e.g. GnRH analogs, aromatase inhibitors etc.). Therefore could be recognized as non-hormonal, a kind of fertility-sparing  therapy. As this might sound confusing for the potential reader we have changed the “a non-hormonal fertility friendly drug” to “a possible new type of pharmacological fertility-sparing treatment method”.

Another example where coherent assembly is missing is the topic of DNA methylation, which is the main idea driving this project. The reader meets this subject on p.9, line 336 (just below Table 2) but then the exact same idea is picked-up again much later, as an entire section on p.13, line 552+. One wonders if six authors just independently gathered literature on their own, then put everything together later with no overall plan.

We deleted some information from the section entitled “Role of TET enzymes in UF development” to only focus on TETs in this fragment.

The section on p.13 entitled “ Demethylating agents and UF development”  were gathered information about known and newly described compounds with the inhibition of methylation enzymes activity.

Minor points

See p.2, line 61: what ‘mechanical forces’ induce uterine fibroids?

Excess of extracellular matrix (ECM), with high amounts of glycosaminoglycans, fibronectins, and, disordered, interstitial collagens result in stiffness of fibroid tissue. These changes generate mechanical stress which is converted into cells into chemical signals and affects gene expression. We added a sentence of explanation into the text.

See p.2, line 70-71: “…mutations are believed to be precursors to chromosomal rearrangements that alter genomic integrity …” Certainly a big insight in 1920, but not sure what this adds in the modern era.

We deleted the sentence suggested by Reviewer as not appropriate.

 See p.4, lines 154-158: “However, DNA methylation status…” This long passage does not appear to be a complete sentence.

We corrected mentioned sentence.

See p.5, lines 237-241: It is unclear how endometrial function or menstrual cyclicity is relevant to myomas, except for cases where fibroids impinge into the uterine cavity. Indeed, if the process-overlap between UF and endometrium is so rare, then this part could be omitted entirely.

We delated the fragment suggested by the Reviewer.

Figure 2 caption: Please define all abbreviations in the figure legend.

We decided to delete Figure 2, as it didn’t contain any relevant information.

The authors may wish to consider developing a summary diagram (alongside Section 5.1, which is data-rich) to illustrate how proposed UF interventions work, either alone or in concert with other therapeutic modes.

We added the diagram (figure 3) where we summarized information from the section entitled “Factors involved in the regulation of TET activity”.

Round 2

Reviewer 2 Report

The suggested modifications appear satisfactory.